# Coagulation abnormalities in children with uncorrected congenital heart defects seen at a teaching hospital in a developing country

Omotola O. Majiyagbe[1,2]*, Adeseye M. Akinsete[1,3], Titilope A. Adeyemo[4], Abideen O. Salako[5], Ekanem N. Ekure[1,3], Christy A. N. Okoromah[1,3]

1 Department of Paediatrics, Lagos University Teaching Hospital, Lagos, Nigeria, 2 Department of Paediatrics, Massey Street Children's Hospital, Lagos Island, Lagos, Nigeria, 3 Department of Paediatrics, College of Medicine of the University of Lagos, Lagos, Nigeria, 4 Department of Pathology, College of Medicine of the University of Lagos, Lagos, Nigeria, 5 Clinical Sciences Department, Nigerian Institute of Medical Research, Lagos, Nigeria

* omotolamajiyagbe@yahoo.com

## Abstract

### Background

Coagulation abnormality is a significant complication and cause of mortality in children with uncorrected congenital heart defects (CHD). The aim of this study was to determine the prevalence of coagulation abnormalities and the associated factors in children with uncorrected CHD.

### Method

A cross sectional study conducted to determine the prevalence of coagulation abnormalities among 70 children with uncorrected CHD aged six months to 17 years and 70 age and sex matched apparently healthy controls. Coagulation abnormalities was determined using complete blood count, prothrombin time, activated partial thromboplastin time and D-dimer assay.

### Results

The prevalence of coagulation abnormalities among children with CHD and controls was 37.1% and 7.1% respectively. Children with Cyanotic CHD had a significantly higher prevalence of coagulation abnormalities compared to children with Acyanotic CHD (57.1% versus 17.1%). Haematocrit and oxygen saturation levels were significantly associated with coagulation abnormalities.

### Conclusion

This study affirms that coagulation abnormalities are frequent in children with uncorrected CHD. Oxygen saturation and haematocrit are risk factors of coagulation abnormalities. Routine coagulation screen is recommended especially in children with cyanotic congenital

---

**Data Availability Statement:** All relevant data are within the paper and its Supporting Information files.

**Funding:** The authors received no specific funding for this work.

**Competing interests:** The authors have declared that no competing interests exist.

heart defects to improve their quality of life and reduce morbidity and mortality while awaiting definitive surgeries.

## Introduction

Globally, congenital heart defect (CHD) defined as a structural and functional defect of the heart present at birth, affecting the heart or adjacent great blood vessels, detected either at birth or later in life, accounts for nearly a third of all major congenital anomalies with an estimated prevalence of 8–12 per 1000 live births [1–3].

In Nigeria, the estimated prevalence varies from 4.6–9.3 per 1000 children [4, 5]. The prevalence of uncorrected CHD, growing to adulthood in Nigeria is unknown, however, in some developed countries like United States, 85% of children with CHD are documented to survive into adulthood since the advent of neonatal repair of complex heart lesions [6]. A multi-center hospital survey in Nigeria by Ekure *et al* [7] in 2017 reported that 84% of the enrolled 1,296 children with CHD still had uncorrected lesions. The delay in corrective surgery was due to late presentation, inadequate infrastructure as well as a high cost of cardiac care. Morbidity from uncorrected CHD may be due toanaemia, infections, malnutrition, congestive cardiac failure, coagulation abnormalities, cerebrovascular accident, bleeding diathesis and pulmonary hypertension, which can have a devastating effect on their quality of life [8].

Coagulation defined as a physiologic process of maintaining haemostasis in the body, can be disrupted in children with CHD leading to functional and structural abnormalities. This is usually due to the chronic state of hypoxia in cyanotic congenital heart defect (CCHD)as well as chronic heart failure in children with acyanotic congenital heart defect (ACHD). These mechanisms lead to compensatory erythrocytosis, hyperviscosity, thrombocytopaenia, platelet aggregation suppression, factor deficiencies, fibrinolysis and thromboembolic phenomenon [9]. In addition, children with cyanotic congenital heart defects have a significant tendency to bleed due to over production of platelet microparticles and shear stress from blood hyperviscosity. Most cases of coagulation abnormalities are mild and asymptomatic, however, severe cases can lead to major complications such as cerebrovascular accidents, limb loss from ischaemia and disseminated intravascular coagulopathy (DIC) which may worsen morbidity; the effects of which may persist even after correction of the heart defects [10].

Coagulation abnormalities can be assessed using the baseline tests for detecting coagulation abnormalities [Prothrombin time (PT), activated partial thromboplastin time (APTT)]in addition to D-dimer assay which has a 95% sensitivity for detecting disseminated intravascular coagulation (DIC) [11]. The baseline test results have been reported to correlate with the newer point of care methods for assessing coagulation abnormalities such as Rotational thromboelastometry (ROTEM), which are expensive and not available in our environment [12].

There are limited data on the prevalence and significance of CHD coagulation abnormalities in developing countries. Studies from developed countries reported a wide prevalence of 19 to 70% [13, 14], thus extrapolating these findings to populations in developing countries might be misleading. This is because the spectrum of routine care and affordability of interventions available in developed countries is mostly unavailable in developing countries like Nigeria. Furthermore, coagulation abnormalities in CCHD have been studied more frequently than ACHD, although acyanotic CHD is commonly seen [15]. Therefore, to improve the quality of care offered to this population until definitive or palliative intervention is obtained, it is important to identify, monitor and treat any coagulation abnormalities to avoid development of irreversible life-limiting end-organ damage.

Thus, the objective of this study was to determine the prevalence of coagulation abnormalities and identify associated risk factors in a cohort of Nigerian children with uncorrected congenital heart defects.

## Materials and methods

It was a cross sectional study conducted at the Lagos University Teaching Hospital (LUTH), Lagos, South-west Nigeria from March 2019 to September 2019, after ethical approval. Ethical approval was obtained from the Lagos University Teaching Hospital Health Research Ethics Committee (LUTHHREC), with assigned number: ADM/DCST/HREC/APP/2387. A written informed consent was obtained from parents or guardians of all study participants with additional informed written assent from children aged seven years and above. The institution is a tertiary hospital that receives referrals from other healthcare facilities in the state and its environs.

Study participants were recruited from the Cardiology clinic, Cardiovascular laboratory and Paediatric outpatient clinics of the hospital using a consecutive sampling method.

### Sample size determination

At the 5% significance and precision level [16], an estimated 70 children with uncorrected congenital heart defects (35 each of ACHD and CCHD) are required to detect a difference in the prevalence of coagulation abnormalities between the control group (20% prevalence) and the cases (70% prevalence) estimated from previous studies [13, 17], at 90% power and adjusting for 10% missing data and unusable samples. Seventy age and sex-matched apparently healthy children without CHD were recruited as controls.

This was calculated using the sample size calculation for difference in two proportions [16].

$$n = \frac{[p_1(1 - p_1) + p_2(1 - p_2)]}{(p_1 - p_2)^2} \times c_{p,power}$$

Where,

$p_1$ = proportion of coagulation abnormalities in uncorrected congenital heart defect, which is estimated as 70% [13].

$p_2$ = proportion of coagulation abnormalities in control group, which is 20% [17]

$c_{p,power}$ = a constant defined by the values chosen for the p-value (0.05) and power of 90%

### Study population

The presence of congenital heart defect was based on the documentary evidence of echocardiography report. Thereafter, consecutive patients with uncorrected CHDs between 6 months to 17 years of age were included in the study while healthy age and sex matched children, who had no clinical evidence of congenital heart defects were recruited as controls. A written informed consent was obtained from parents or guardians of all study participants with additional informed written assent from children aged seven years and above.

Children with trans-catheter or surgically palliated or corrected congenital heart defects were excluded. Also, children previously diagnosed with genetic syndromes, chronic renal failure, chronic liver disease, malignancies, HIV infection, sickle cell anaemia were excluded from the study. Children who had been on non-steroidal anti-inflammatory, anti-platelet and anticoagulant therapies in the preceding two weeks, and those with family history of bleeding

disorders such as haemophilia were also excluded. Clinical examination, including oxygen saturation with pulse oximetry and anthropometry were carried out and a pre-tested proforma (S1 Appendix) was also used to obtain biodata, clinical information, drug history and socio-demographic characteristics (age, gender, socioeconomic class which included occupational status and highest level of education attained by each parent and/or guardian using Oyedeji classification of social class) [18]. Heart failure was classified using the Modified Ross Heart Failure Classification for Children. The medications the children with congenital heart defects were on at time of evaluation were frusemide, spironolactone and propranolol.

Prothrombin time (PT) and APTT values greater than 2SD of the mean values were reported as prolonged, as recommended by the Clinical & Laboratory Standard Institute (CLSI) on defining, establishing and verifying reference intervals in the clinical laboratory [19].

## Blood sample collection

Four milliliters of venous blood was withdrawn aseptically from each participant; two milliliters was added into a marked 3.2% citrated bottle for the clotting profile analysis (so as to maintain a standard 9:1 blood to anticoagulant ratio), while two milliliters was collected into an ethylene diamine tetra-acetic acid (EDTA) bottle for evaluation of haematocrit and platelet count. For cyanotic patients with known haematocrit levels above 55%, the haematocrit value was first checked and the amount of anticoagulant in the collection tube was adjusted according to the CLSI guidelines [19] using the formula,

$$C = 1.85 \times 10^{-3}(100 - h) \times V$$

Where, C = volume of 3.2% sodium citrate in ml, h = haematocrit and V = Volume of whole blood in mls that was added to the anticoagulant.

Blood samples were transported in a cooler box (+4 to +8 $^0$C) in an upright position on the tube racks to the Central Research Laboratory of the College of Medicine, University of Lagos (CMUL), Lagos. Samples that were deemed insufficient, with blood clots or evidence of haemolysis were rejected from analysis. Within 4 hours of collection, the samples were centrifuged at 1500 rpm for 15mins, the resultant platelet poor plasma was extracted using pipettes into cryo bottles in three aliquots of 0.5mls and stored at -70 $^0$C at the CMUL Central Research Laboratory for a month to maintain potency until analysis.

## Laboratory analysis

Complete blood count was analysed using a Mindray BC-3200 Automated Haematology Analyzer (Shenzhen, China). Prothrombin time and activated partial thromboplastin time were determined using the coagulometer, a CA-101 (Sysmex, India).

D-dimer was assayed by enzyme linked immunosorbent assay (ELISA) using the Human D-dimer ELISA microwells kit (Bioassay Technology Laboratory, China) designed for quantitative detection of d-dimer in human serum or plasma.

Coagulation abnormality was defined as thrombocytopaenia, prolonged prothrombin time and/or activated partial thromboplastin time and elevated D-dimer assay levels.

## Follow up of study participants

Laboratory results were made available to study participants. This was done mainly by telephone calls and distributing personal hard copies of the laboratory results to the parents/guardian during their follow up appointments. For the children with abnormal but asymptomatic coagulation tests, results were communicated to their managing physicians, to be co-

managed with the Haematology unit of LUTH through short clinic appointments with a recommendation for proactive vigilance for a bleeding diathesis or thrombotic event or both. The caregivers of children with uncorrected CHD were also counseled on the need for prompt surgical correction of the cardiac defects.

### Data analysis

Statistical analysis was performed using Statistical Package for Social Sciences (SPSS) (version 21) Armonk, NY:IBM Corp.

Quantitative data was tested for normality using Shapiro-Wilk test. Normally distributed data was summarized using mean and standard deviation, while skewed data was presented as median and interquartile range. Conversely, categorical variables (demographic, anthropometry and laboratory findings) were summarized using frequencies and proportions. Comparison of the prevalence of coagulation abnormalities between the subject group and control group was done using the Chi square test. Comparison of normally distributed continuous data (laboratory findings) between the cyanotic and acyanotic groups was tested using student t test while similar comparison between skewed data were made using Mann Whitney U test. Comparison of categorical variables between cyanotic and acyanotic groups was done using the Chi-square test. Variables that were statistically significantly associated with coagulation abnormalities were subjected to further statistical test using multiple logistic regression to determine the independent predictors of coagulation abnormalities. For all statistical analyses, p value less than 0.05 was considered statistically significant.

## Results

A total of 144 children were recruited for the study. Four of them were excluded from data analysis on account of inability to run the laboratory tests due to clotted samples.

### Characteristics of CHD subjects and the controls

A total of one hundred and forty children were enrolled in the study: seventy children each in the uncorrected congenital heart defects and apparently healthy aged and sex matched control groups.

Table 1 shows a summary of the demography and anthropometric findings in CHD subjects and controls. The study population ages ranged from 6 months to 15 years with majority in the 1–5 years group. The median (IQR) age of the two groups was comparable 36.0 (18.0–84.0) months in the CHD and 48.0 (20.0–87.0) months in the healthy control (p-value = 0.985). The male-female proportion was 42.9% and 57.1%. Most of the caregivers of the study population were in the middle socioeconomic class. The median weight, height and body mass index were lower for CHD subjects compared to controls. Children with CHD were significantly stunted and wasted compared to the healthy controls. Twenty-nine (41.4%) and 26 (37.2%) of the children with uncorrected CHDs were wasted and stunted respectively compared to 11 (15.7%) and 10 (14.3%) of the healthy controls who were wasted and stunted, p = <0.001.

### Characteristics of children with CHD

For children with acyanotic congenital heart defects: Isolated VSD 14 (40%), Isolated ASD 6 (17.1%) and Isolated PDA 3 (8.5%) were the most common types, while Tetralogy of Fallot 20 (57.1%) and Truncus arteriosus 4 (11.4%) were the commonest CCHD. Heart failure was diagnosed in 32 of the 70 CHD children, of which 87.5% of them had ACHD (Modified Ross Heart failure classification) as shown in Table 2.

**Table 1. Socio-demographic and anthropometry of study population.**

| Variable N (%) | CCHD n = 35 | ACHD n = 35 | Controls n = 70 | Test Statistic | P–value |
|---|---|---|---|---|---|
| **Age,** median (IQR), years | 3.0 (1.8 to 7.0) | 4.0 (1.4 to 8.0) | 4.0 (1.7 to 7.3) | +0.079 | 0.961 |
| **Age group** | | | | | |
| 6 months– 1 year | 4 (11.4) | 4 (11.4) | 8(11.4) | *0.076 | 1.000 |
| 1 to <5 years | 18 (48.6) | 16 (45.7) | 34(48.6) | | |
| 5 to <10 years | 8 (22.9) | 11 (31.4) | 19(27.1) | | |
| ≥10 years | 5 (14.3) | 4 (11.4) | 9(12.9) | | |
| **Sex** | | | | | |
| Females | 18 (51.4) | 22 (62.9) | 40(57.1) | $\chi^2 = 0.933$ | 0.627 |
| Males | 17 (48.6) | 13 (37.1) | 30(42.9) | | |
| **Socioeconomic class** | | | | | |
| Low | 6 (17.1) | 5 (14.3) | 4 (5.7) | $\chi^2 = 6.764$ | 0.149 |
| Middle | 17 (48.6) | 18 (51.4) | 29 (41.4) | | |
| High | 12 (34.3) | 12 (34.3) | 37 (52.9) | | |
| **Anthropometry** | | | | | |
| BMI for age, median (IQR) | -1.5(-3.2 to -0.7) | -1.9(-2.6 to -0.7) | -0.4(-1.0 to -0.9) | +27.274 | <0.001# |
| Height for age, median (IQR) | -1.4(-3.3 to -0.5) | -1.6(-2.3 to -0.4) | 0 (-1.3 to 0.8) | +22.277 | <0.001# |
| Weight for age, median (IQR) | -2.0(-3.2 to 1.3) | -1.8(-2.9 to -1.1) | -0.4 (-0.9 to 0.4) | +46.580 | <0.001# |
| **BMI for age,** *n (%)* | | | | | |
| Normal | 20 (57.1) | 21 (60.0) | 52 (74.3) | | |
| Wasted | 5 (14.3) | 8 (22.9) | 10 (14.3) | *24.231 | <0.001# |
| Severely wasted | 10 (28.6) | 6 (17.1) | 1 (1.4) | | |
| Overweight | 0 (0.0) | 0 (0.0) | 7 (10) | | |
| **Height for age,** *n (%)* | | | | | |
| Normal | 22 (62.9) | 22 (62.9) | 55 (78.6) | | |
| Stunted | 4 (11.4) | 9 (25.7) | 9 (12.9) | *20.871 | <0.001# |
| Severely stunted | 9 (25.7) | 4 (11.4) | 1 (1.4) | | |
| Tall | 0 (0.0) | 0 (0.0) | 5 (7.1) | | |
| **Oxygen Saturation**, % | 77.2 ± 9.8 | 94.8 ± 5.2 | 98.4 ± 0.9 | * 174.654 | <0.001# |
| Range | 60–95% | 77–99 | 95–100 | | |

+Kruskal-Wallis Test

*Fisher's exact test

#*p* value statistically significant

## Prevalence of coagulation abnormalities in children with CHD and controls

The proportion of uncorrected CHD cohort with coagulation abnormalities, defined as thrombocytopaenia, prolonged prothrombin time and/or activated partial thromboplastin time and elevated D-dimer assay was 26 (37.1%), which was much higher than 5 (7.1%) in apparently healthy controls as shown in Fig 1.

## Comparison of coagulation profile in children with CHD and controls

Table 3 shows the mean values of coagulation parameters in CHD sub-group subjects and controls. Children with uncorrected CHD had prolonged APTT and PT compared with controls. The difference in mean prothrombin time among CHD sub-group and controls was

**Table 2. Distribution of congenital heart defects.**

| Characteristics of CHD | n (%) |
|---|---|
| **Acyanotic CHD** | |
| Isolated VSD | 14 (40) |
| Isolated PDA | 3 (8.5) |
| Isolated ASD | 6 (17.1) |
| VSD+PDA | 1 (2.9) |
| VSD+PS | 1 (2.9) |
| AVSD | 4 (11.4) |
| VSD+ASD | 3 (8.5) |
| ASD+PS | 1 (2.9) |
| Bicuspid aortic valve | 1 (2.9) |
| Coarctation of the aorta | 1 (2.9) |
| **Total** | **35 (50)** |
| **Cyanotic CHD** | |
| TOF | 20 (57.1) |
| DORV | 2 (5.7) |
| DILV | 1 (2.9) |
| Truncus arteriosus | 4 (11.4) |
| PA | 2 (5.7) |
| Tricuspid atresia | 2 (5.7) |
| Ebstein anomaly | 1 (2.9) |
| TGA | 2 (5.7) |
| HLHS | 1 (2.9) |
| **Total** | **35 (50)** |
| **Heart failure Present** | |
| **ACHD (35)** | 28 |
| **CCHD (35)** | 4 |

*VSD: ventricular septal defect; PDA: patent ductus arteriosus; ASD: atrial septal defect; PS: pulmonary stenosis; AVSD: atrioventricular septal defect; TOF: tetralogy of Fallot; DORV: double outlet right ventricle; DILV: double inlet left ventricle TA: Truncus arteriosus; PA: pulmonary atresia, TGA: Transposition of Great arteries, HLHS: hypoplastic left heart syndrome.

statistically significant (p-value <0.05). The median platelet count of the children with congenital heart defects was much lower than the median of the controls. However, there was no significant difference in the D-dimer levels between children with CHD and controls.

## Prevalence of coagulation abnormalities in children with acyanotic and cyanotic congenital heart defects

Coagulation abnormalities were more common in the cyanotic CHD group compared to the acyanotic group [20 (57.1%) versus 6 (17.1%)] and controls with a p value of <0.001. The proportion of coagulation abnormalities in the two subgroups is as shown in Table 4

Six children with ACHD had coagulation abnormalities: 5 had abnormal results in only one test while 1 had in two tests.

Twenty children with CCHD had coagulation abnormalities: 12 had abnormal results in one test while 8 had in two or more tests.

Children with uncorrected CHD who had coagulation abnormalities had lower mean oxygen saturations, as shown in Table 5.

RESULTS

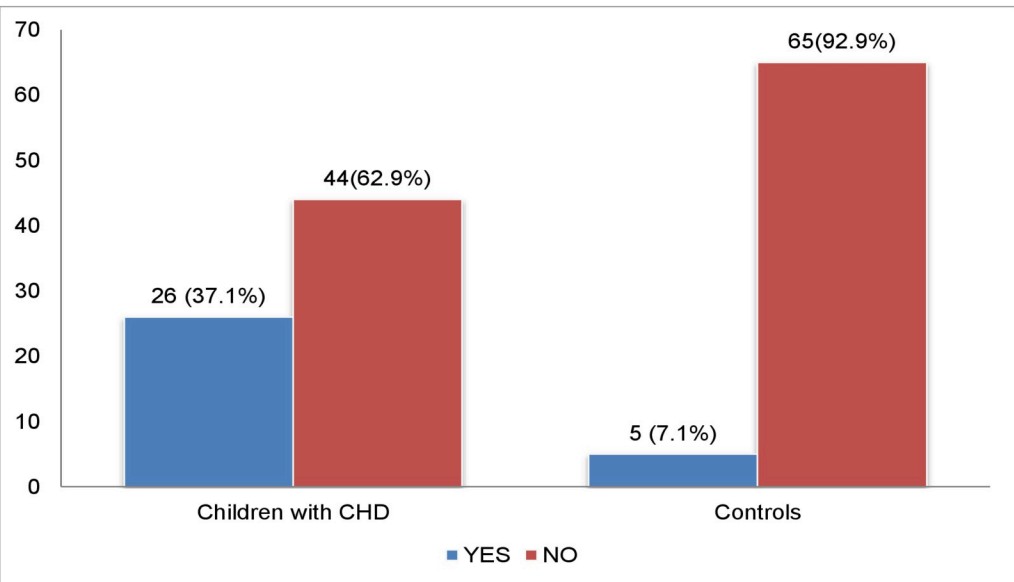

**Fig 1. Prevalence of coagulation abnormalities in children with uncorrected CHD and controls.**

## Factors associated with coagulation abnormalities in children with CHD

There was no significant association between the socio-demographic characteristics (age, sex, and socioeconomic class), nutritional status (height for age, BMI)and the prevalence of coagulation abnormalities in children with uncorrected congenital heart defects. Coagulation abnormalities were frequent in CHD subjects that were less than 5 years of age, though it was not statistically significant. CHD subjects with coagulation abnormalities had higher mean values of haematocrit and lower mean values of oxygen saturation levels as shown in Table 6.

Coagulation abnormalities were noted in CHD children with heart failure though it was not statistically significant, as shown in Table 7

Table 8 showed that socio-demographic characteristics (age, sex) and anthropometry were not independent predictors of coagulation abnormalities. Coagulation abnormalities was 6 times [6.444 (2.135–19.456)] more likely to occur in children with CCHD compared with children with ACHD.

**Table 3. Comparison of coagulation profile between CHD subjects and controls.**

| Variable N (%) | CCHD n = 35 | ACHD n = 35 | Controls n = 70 | Test Statistic | P–value |
|---|---|---|---|---|---|
| **Median (IQR) platelet count** (x$10^9$ cells/mm$^3$) | 166.0(129–275) | 290(217–403) | 356(274.5–443) | +33.805 | <0.001# |
| **Prothrombin time,** mean ± SD, seconds | 16.2 ± 4.1 | 13.6 ± 2.3 | 13.0 ± 1.7 | *15.963 | <0.001# |
| **International Normalized Ratio** mean ± SD | 1.2 ± 0.3 | 1.0 ± 0.2 | 1.0 ± 0.1 | *20.016 | <0.001# |
| **Activated partial thromboplastin time,** mean ± SD, seconds | 45.1 ± 13.0 | 43.8 ± 8.1 | 39.7 ± 7.3 | *4.792 | 0.010 |
| **D-dimer** Median (IQR) (pg/ml) | 0.9(0.7 to 0.9) | 1.0(0.8 to 1.9) | 1.2(0.8 to 2.1) | +2.912 | 0.233 |

+Kruskal-Wallis Test

*One way ANOVA

#p value statistically significant

**Table 4. Coagulation abnormalities in children with ACHD and CCHD.**

| Variable N (%) | ACHD n = 35 | CCHD n = 35 | Controls n = 70 | Test statistic $\chi^2$ | p-value |
|---|---|---|---|---|---|
| **Coagulation abnormality** | 6 (17.1) | 20 (57.1) | 5 (7.1) | 34.513 | <0.001 |
| **Thrombocytopaenia** | 0(0.0) | 8(22.9) | 2(2.9) | *13.772 | <0.001 |
| **Prolonged PT/INR** | 2(5.7) | 11(31.4) | 1(1.4) | *20.390 | <0.001 |
| **Prolonged APTT** | 2(5.7) | 9(25.7) | 0(0.0) | *19.141 | <0.001 |
| **Elevated D-dimer(pg/ml)** | 3(8.6) | 3(8.6) | 2(2.9) | 2.474 | 0.385 |

$\chi^2$:Chi-square

*Fisher's exact

#*p* value statistically significant

Table 9 showed that haematocrit level and oxygen saturation had a statistically significant association on levels of platelet count, prothrombin time and INR. The model shows that haematocrit value and oxygen saturation have stronger effects on the platelet count and prothrombin time/INR.

Pearson's correlation was carried out to demonstrate relationship between coagulation abnormalities and factors such as age, anthropometry, oxygen saturation and haematocrit.

Table 10 shows that haematocrit value and oxygen saturation had stronger effects on the platelet count and prothrombin time/INR. There was moderate negative correlation between haematocrit and platelet count (r = -0.414, p = 0.000) and a moderate positive correlation between the haematocrit level of children with CHD and prothrombin time/ INR (r = +0.416, p = 0.000). There was also a positive moderate correlation between the oxygen saturation level of children with CHD and platelet count (r = +0.443, p = 0.000). There was no correlation between age of patient, anthropometry and coagulation abnormalities.

## Discussion

This study determined the prevalence of coagulation abnormalities (thrombocytopaenia, prolonged clotting profile and elevated d-dimer levels) in children with uncorrected congenital heart defects and reported a prevalence of 37.1%, compared to 7.2% in the healthy cohort. The mean clotting profile of children with uncorrected CHD was significantly prolonged. These findings in our study can be attributed to the uncorrected congenital heart defect, which has been reported to cause a chronic consumption coagulopathy [8].

Our prevalence was higher than that reported by Colon et al in the United States among a mixed cohort of adults and children, some of whom had corrected heart defects [14]. This is

**Table 5. Oxygen saturation levels in participants with uncorrected CHD.**

| | Mean | Test Statistic* | P-Value |
|---|---|---|---|
| **With coagulation abnormalities** | | | |
| ACHD | 92.0 ± 7.0 | F = 14.202 | <0.001 |
| CCHD | 76.0 ± 9.6 | | |
| **Without coagulation abnormalities** | | | |
| ACHD | 95.4 ± 4.8 | | |
| CCHD | 78.8 ± 10.1 | | |

*Between Groups ANOVAs

**Table 6. Association between socio demographic characteristics, nutritional status and the prevalence of coagulation abnormalities in children with CHD.**

| Variables (n = 70) | Coagulation abnormalities | | Test statistics | P-value |
|---|---|---|---|---|
| | Yes (n = 26) | No (n = 44) | | |
| **Age group** | | | | |
| 6 months to <1 year | 3(11.5) | 5(11.4) | | |
| 1 year to <5 years | 13(50.0) | 21(47.7) | $\chi^2$ = 1.076 | 0.783 |
| 5 years to <10years | 8(30.8) | 11(25.0) | | |
| ≥10 years | 2(7.7) | 7(15.9) | | |
| **Sex** | | | | |
| Male | 11(42.3) | 19(43.2) | $\chi^2$ = 0.193 | 0.660 |
| Female | 15(57.7) | 25(56.8) | | |
| **Height for age** | | | | |
| Normal | 16(61.6) | 28(63.6) | | 0.985 |
| Stunted | 5(19.2) | 8(18.2) | $\chi^2$ = 0.031 | |
| Severely stunted | 5(19.2) | 8(18.2) | | |
| **BMI for age** | | | | |
| Normal | 13(50.0) | 28(63.6) | $\chi^2$ = 1.661 | 0.436 |
| Wasted | 5(19.2) | 8(18.2) | | |
| **Oxygen saturation in room air (mean ± SD)** | 79.6 ± 11.3 | 89.8 ± 10.6 | t = 14.202 | 0.000 |
| **Haematocrit (mean ± SD)** | 46.5 ± 14.5 | 38.8 ± 10.7 | t = 6.541 | 0.014 |

$\chi^2$:Chi-square. t = Student t-test

possibly due to the cohort of children with uncorrected CHDs in the current study, which reflects the pool of children with CHDs in sub-Saharan Africa where cardiac intervention is delayed due to high cost of surgery and inadequate manpower. In comparison, our result was lower than reports from the Asia sub-regions, which were in small patient populations [13, 20]. These investigators proposed that their cohorts had severe cardiac defects. Nevertheless, the relatively high proportion of coagulation abnormalities between CHD cohort in this current study compared to the healthy population further highlights that chronic disseminated intravascular coagulopathy occur in children with CHD. The prevalence among the apparently healthy population in this current study was comparable to the report by Alzahrani et al [21] in Saudi Arabia. It was however lower than figures reported by Acosta et al [17] in the US and Onakoya [22] in South-west Nigeria. The low rates in the current study might be linked to the exclusion of children with bleeding disorders as well as the study design utilized. It may then be assumed that coagulation disorders occur even among apparently healthy children. Viral infections have been implicated in the prevalence of coagulation defects in apparently healthy cohorts. This is due to the elaboration of cytokines that alter the balance between pro and anti-coagulation factors. Thus, pre surgical coagulation screening for all children should be advocated regardless of absence of a family history of bleeding disorders.

**Table 7. Association between coagulation abnormalities & heart failure in children with uncorrected congenital heart defects.**

| | | Coagulation profile | | | | |
|---|---|---|---|---|---|---|
| | | Abnormal | Normal | Total | Test Statistic | P-Value |
| Heart Failure | Yes % | 7 (21.9) | 25 (78.1) | 32 (45.7) | $\chi^2$ = 5.886 | 0.014 |
| | No % | 19 (50.0) | 19 (50.0) | 38 (54.3) | | |

$\chi^2$: Chi square

**Table 8. Independent variables associated with coagulation abnormalities.**

| Variable | Adjusted odds ratio | P value | 95% CI | |
|---|---|---|---|---|
| | | | Lower bound | Upper bound |
| | | | Coagulation profile | |
| **Age group** | | | | |
| 6 months– 1 year (vs. ≥10 years) | 2.100 | 0.494 | 0.251 | 17.594 |
| 1 to <5 years (vs. ≥10 years) | 2.167 | 0.377 | 0.389 | 12.063 |
| 5 to <10 years (vs. ≥10 years) | 2.545 | 0.313 | 0.414 | 15.652 |
| **Sex** | | | | |
| Male vs. Female | 1.254 | 0.660 | 0.469 | 3.309 |
| **Cardiac defects** | | | | |
| Cyanotic CHD vs Acyanotic | 6.444 | 0.001 | 2.135 | 19.456 |
| **Height for age** | | | | |
| Normal (vs. severely stunted) | 1.000 | 1.0000 | 0.206 | 4.856 |
| Stunted (vs. severely stunted) | 0.914 | 0.890 | 0.255 | 3.272 |
| **BMI for age** | | | | |
| Normal (vs. severely wasted) | 0.464 | 0.203 | 0.143 | 1.511 |
| Wasted (vs. severely wasted) | 0.625 | 0.535 | 0.141 | 2.763 |

The prevalence of coagulation abnormalities in children with CCHD was significantly higher than the ACHD cohort, which is similar to earlier published reports [14]. In our population of children with CHD, slightly less than a quarter of the CCHD arm had thrombocytopenia with no thrombocytopenia reported among the ACHD arm. This is similar to a report in Sweden which had a prevalence of 15% among a cohort of children with CCHD, and the authors attributed the thrombocytopenia to hypoxic inhibition of platelet production [23]. This was also corroborated by other studies from Europe and North America [14, 24]. In our study, all the children who had thrombocytopenia had oxygen saturation less than 85%. This is

**Table 9. Multivariate logistic analysis of significant variables and coagulation abnormalities.**

| Variable[a] | B | Standardized Beta-coefficient | SE | p-value | 95% CI | |
|---|---|---|---|---|---|---|
| | | | | | Lower bound | Upper bound |
| | | | | Platelet count (cell/mm3) | | |
| Haematocrit (%) | -5073.1 | -0.408 | 1391.0 | 0.001 | -7850.4 | -2295.8 |
| Oxygen saturation | 5886.7 | 0.438 | 1477.1 | 0.000 | 2935.8 | 8837.6 |
| | | | | Prothrombin time (secs) | | |
| Haematocrit (%) | 0.117 | 0.426 | 0.030 | 0.000 | .057 | .177 |
| Oxygen saturation | -0.085 | -0.286 | 0.034 | 0.016 | -0.153 | -0.016 |
| | | | | International normalised ratio | | |
| Haematocrit (%) | 0.011 | 0.480 | 0.002 | 0.000 | 0.006 | 0.015 |
| Oxygen saturation | -0.007 | -0.300 | 0.003 | 0.011 | -0.013 | -0.002 |
| | | | | APTT | | |
| Haematocrit (%) | 0.152 | 0.180 | 0.102 | 0.142 | -0.052 | 0.355 |
| Oxygen saturation | 0.004 | 0.005 | 0.112 | 0.969 | -0.219 | 0.228 |
| | | | | D-Dimer (pg/ml) | | |
| Haematocrit (%) | -0.005 | -0.076 | 0.008 | 0.527 | -0.021 | 0.011 |
| Oxygen saturation | 0.000 | -0.004 | 0.009 | 0.975 | -0.018 | 0.017 |

[a]Adjusted for socioeconomic class and nutritional status

**Table 10. Correlation between coagulation parameters and some independent factors.**

| Variables (n = 70) | Platelet count | Prothrombin time | Partial thromboplastin time | International Normalised ratio | D-dimer |
|---|---|---|---|---|---|
| Age | r = -0.215 | r = 0.099 | r = -0.068 | r = 0.199 | r = -0.324 |
| | p = 0.074 | p = 0.415 | p = 0.577 | p = 0.099 | p = 0.006 |
| Weight for age | r = -0.122 | r = -0.143 | r = -0.015 | r = -0.146 | r = -0.031 |
| | p = 0.340 | p = 0.264 | p = 0.908 | p = 0.254 | p = 0.811 |
| Height for age | r = -0.063 | r = -0.076 | r = -0.041 | r = -0.087 | r = -0.084 |
| | p = 0.604 | p = 0.531 | p = 0.735 | p = 0.473 | p = 0.489 |
| BMI for age | r = -0.104 | r = -0.202 | r = 0.044 | r = -0.199 | r = 0.000 |
| | p = 0.392 | p = 0.093 | p = 0.718 | p = 0.098 | p = 1.000 |
| Oxygen saturation | r = 0.443 | r = -0.285 | r = 0.000 | r = -0.298 | r = -0.024 |
| | p = 0.000 | p = 0.017 | p = 0.997 | p = 0.012 | p = 0.841 |
| Haematocrit level | r = -0.414 | r = 0.416 | r = 0.187 | r = 0.471 | r = -0.066 |
| | p = 0.000 | p = 0.000 | p = 0.122 | p = 0.000 | p = 0.586 |

similar to the North Africa study [25]. Children with CCHD have been reported to have decreased platelet counts due to diverse mechanisms such as increased platelet destruction, aggregation and activation as well as reduced platelet production due to the right to left shunts, which deliver megakaryocytes into the systemic arterial circulation bypassing the lungs where megakaryocytic cytoplasm is fragmented into platelets [26, 27].

Similar to earlier reports, PT and APTT were the most deranged coagulation screening tests in this study [13, 14]. Prolonged PT/INR and APTT were significantly higher in children with CCHD compared to the ACHD cohort. A high prevalence (35.5% and 37.5%) of prolonged PT/INR in CCHD has also been reported among children with CCHD in East Africa [25, 28]. A higher rate of prolonged clotting factors was however reported from the United States among children aged 3 to 19 months with CCHD [29]. This was probably due to the predominance of infants in their study group, which may have partly accounted for the higher rates, as infants are known to have physiologic maturational delay in haemostasis [30]. Majority of the children with CHD who had prolonged PT, APTT in our study had oxygen saturation less than 90% (all the CCHD and one patient with atrioventricular septal defect who had severe pulmonary hypertension). Hypoxia has been reported to induce prothrombotic state by increasing the expression of coagulation factors [31]. It also causes vascular remodeling by release of hypoxia inducing factors and Weibel-Palade bodies in the endothelial cells and these stimulate thrombosis. Severe pulmonary hypertension results in systemic oxygen desaturation and erythrocytosis. This subsequently leads to abnormal endothelial function and decreased plasma thrombomodulin, with a shift to a procoagulant state [32].

Elevated D-dimer level was seen equally in both ACHD and CCHD groups in the current study. A study in Nairobi however reported a significantly higher rate in the CCHD cohort [28]. Children with CCHD are at a higher risk of chronic thrombus formation due to hypoxic induced polycythaemia while, the chronic heart failure in ACHD causes cardiac dilatation, leading to reduced blood perfusion and possibly blood stasis with consequent tendency to thrombus formation.

Coagulation abnormalities were found to be highest in under fives, compared to the older age group. The difference was however not statistically significant. This finding is similar to the US study [29], who also found more coagulation abnormalities in participants less than 5 years. This suggests that this complication occur at an early age.

Furthermore in the current study, there was no statistically significant relationship between coagulation abnormality and nutritional status.

Similar to published works [13, 14], our study showed a weak and negative correlation between oxygen saturation level and coagulation abnormalities. This relationship suggests that hypoxia contributes significantly to the initiation and sustenance of coagulation abnormalities as it leads to endothelial injury, and exposure of the sub-endothelial space to blood platelets, with subsequent activation of the coagulation cascade [31]. There is thus a state of low grade or subclinical DIC in children with CCHD and ACHD with severe pulmonary hypertension due to the chronic hypoxic state, as evidenced by the alterations in coagulation profiles in our results [33]. There was however no signs of haemorrhage in all our patients. Hypoxia also results in hepatic hypo-perfusion, leading to hepatic ischaemia and reduced production of clotting factors [34, 35].

Authors have reported significantly higher levels of plasma haemostatic markers such as D-dimer, fibrinogen, platelet factor 4 in cases of heart failure. During congestive heart failure, there is a higher level of cytokines, tumor necrotic factor (TNF), interleukin 6 (IL6) and these are important in maintenance and progression of prothrombotic state since they promote inflammation and angiogenesis, as well as enhance procoagulant properties of blood by activation of tissue factors. Furthermore, the acute phase protein synthesis becomes more intensive in the liver due to the high level of cytokines, and there is insufficiency of amino acids required for protein synthesis [36]. Chronic heart failure leads to increased sympathetic activity, which contribute to redistribution of blood away from the splanchnic circulation.

Furthermore, the correlation between haematocrit level, thrombocytopaenia and prolonged prothrombin time though weak, was statistically significant and in concordance with previous studies in Japan and United States of America [24, 32]. This affirms the role of polycythaemia and platelet dysfunction in the promotion of abnormal coagulation cascade in uncorrected congenital heart defects. Elevated haemoglobin results in hyper-viscosity with resultant increased endothelial dysfunction, platelet activation and subsequently accelerated activation of the clotting cascade [37, 38].

The limitation in the present study was inability to use point of care testing for coagulation analysis, which is a more sensitive diagnostic test for coagulation analysis as it uses whole blood sample and better represent actual clotting function.

## Conclusion

It is evident that coagulation abnormalities are more prevalent among children with uncorrected cyanotic CHD when compared to those with acyanotic and apparently normal children. Oxygen saturation and haematocrit are factors associated with these abnormalities. There is however need for further evaluation and possibly a multi-center study to correlate these findings so as to advise institutions on standard clinical care and therapy while the children await palliative or definitive cardiac surgeries. There is thus a need for periodic evaluation for coagulation abnormalities in children with uncorrected congenital heart defects so as to institute therapy when needed to improve their quality of life, and disease burden while they await corrective or palliative surgeries.

We recommend screening for coagulation abnormalities as part of routine management of children with uncorrected CHD.

We also recommend early corrective cardiac interventions for the children with uncorrected CHD so as to improve oxygenation, limit chronic heart failures, prevent development of severe pulmonary arterial hypertension, and development of uncompensated DIC.

## Supporting information

**S1 Appendix. Proforma.**
(DOCX)

**S2 Appendix. Study data.**
(XLS)

**S1 File.**
(PDF)

## Acknowledgments

My sincere gratitude goes to all the children and the parents who participated in this study.

## Author Contributions

**Conceptualization:** Omotola O. Majiyagbe.

**Data curation:** Omotola O. Majiyagbe.

**Formal analysis:** Omotola O. Majiyagbe.

**Investigation:** Omotola O. Majiyagbe.

**Methodology:** Omotola O. Majiyagbe, Adeseye M. Akinsete, Titilope A. Adeyemo, Abideen O. Salako, Ekanem N. Ekure.

**Project administration:** Omotola O. Majiyagbe, Adeseye M. Akinsete, Titilope A. Adeyemo, Abideen O. Salako, Ekanem N. Ekure.

**Resources:** Omotola O. Majiyagbe, Adeseye M. Akinsete, Titilope A. Adeyemo, Abideen O. Salako, Ekanem N. Ekure.

**Supervision:** Adeseye M. Akinsete, Titilope A. Adeyemo, Abideen O. Salako, Ekanem N. Ekure, Christy A. N. Okoromah.

**Validation:** Titilope A. Adeyemo.

**Writing – original draft:** Omotola O. Majiyagbe, Ekanem N. Ekure.

**Writing – review & editing:** Omotola O. Majiyagbe, Adeseye M. Akinsete, Titilope A. Adeyemo, Abideen O. Salako, Ekanem N. Ekure, Christy A. N. Okoromah.

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
