## [Decision Letter · Decision Letter 0]

16 Mar 2022

PONE-D-22-02924Coagulation abnormalities in children with uncorrected congenital heart defects seen at a teaching hospital in a developing countryPLOS ONE

Dear Dr. Majiyagbe, 

Thank you for submitting your manuscript to PLOS ONE. After careful consideration, we feel that it has merit but does not fully meet PLOS ONE’s publication criteria as it currently stands. Therefore, we invite you to submit a revised version of the manuscript that addresses the points raised during the review process.

ACADEMIC EDITOR: The paper is well written but, to be accepted for publication, should be able to answer the queries issued by the reviewers.

To ensure the Editor and Reviewers will be able to recommend that your revised manuscript is accepted, please pay careful attention to each of the comments that have been pasted underneath this email. This way we can avoid future rounds of clarifications and revisions, moving swiftly to a decision. 

Please submit your revised manuscript within 60 days. If you will need more time than this to complete your revisions, please reply to this message or contact the journal office at plosone@plos.org. Please include the following items when submitting your revised manuscript:A rebuttal letter that responds to each point raised by the academic editor and reviewer(s). You should upload this letter as a separate file labeled 'Response to Reviewers'.A marked-up copy of your manuscript that highlights changes made to the original version. You should upload this as a separate file labeled 'Revised Manuscript with Track Changes'.An unmarked version of your revised paper without tracked changes. You should upload this as a separate file labeled 'Manuscript'.

We look forward to receiving your revised manuscript.

Kind regards,

Emanuele Bobbio, M.D.

Academic Editor

PLOS ONE

Reviewers' comments:

Reviewer's Responses to Questions

**Comments to the Author**

1. Is the manuscript technically sound, and do the data support the conclusions?

Reviewer #1: Partly

Reviewer #2: Partly

Reviewer #3: Yes

2. Has the statistical analysis been performed appropriately and rigorously? 

Reviewer #1: Yes

Reviewer #2: I Don't Know

Reviewer #3: Yes

3. Have the authors made all data underlying the findings in their manuscript fully available?

Reviewer #1: Yes

Reviewer #2: Yes

Reviewer #3: Yes

4. Is the manuscript presented in an intelligible fashion and written in standard English?

Reviewer #1: Yes

Reviewer #2: Yes

Reviewer #3: Yes

5. Review Comments to the Author

Reviewer #1: 1. This manuscript reviews 70 cases of uncorrected congenital heart disease in patients 6 months to 17 years of age to establish the prevalence of coagulation defects looking at very basic parameters including CBC, PT, PTT and D-dimer. The study found a significantly elevated incidence of 37% when compared to a control population of 7% abnormalities.

2. The case:control ratio is only 1:1 which limits the reliability of the results, particularly in view of the fact that the controls had a relatively high and unexpected incidence of coagulation abnormalities. A ratio of at least 1:2 would be preferred.

3. The abbreviations CCHD and ACHD are used early in the article and not defined. It becomes apparent with further reading that it does refer to acyanotic and cyanotic heart disease, but it would be less confusion to define this earlier.

4. The details of how the blood samples on pages 9 and 10 were collected are somewhat excessive and could be condensed considerably. It is important to stress the different amount of coagulant required for patients with > 55% hematocrit.

5. There is variability in referring to coagulation abnormalities in the singular or pleural form and should be uniformly made in the pleural form, that is "abnormalities".

6. Many previous studies have shown that the major problems with coagulation occur with cyanotic congenital heart disease. This seems to be supported in this review which shows cyanotic lesions had a 37% incidence of abnormalities compared to a cyanotic lesions of 17%. However, most of the data discussion lumps acyanotic and cyanotic heart disease together and it would be important, even though the numbers are relatively small to break down the difference in these groups more clearly particularly in Table 1,3 and 4. It would also be important to include the pulse ox values in the acyanotic and cyanotic heart disease groups in addition to controls on Table 1. Including specific pulse ox data is critical since this is inherent in their conclusions of hypoxia being a primary factor in producing coagulation abnormalities.

7. In the discussion the authors describe the abnormalities of coagulation to 2 major factors, hypoxia and congestive heart failure. This speculation is intuitively correct, but should be expanded in more detail to include reference to low-grade DIC that is likely present with cyanotic patients. The link between congestive heart failure and probable hepatic hypoproduction of clotting factors should also be more clearly discussed.

Reviewer #2: Thank you for the opportunity to review this interesting paper about coagulation disorders in patients with congenital, native heart defects from Nigeria. The paper is very well written and it is a pleasure to read the paper.

However, there are some major concerns which have to be addressed:

1. The objective of the study was to determine the prevalence of coagulation abnormality in paediatric uncorrected CHD and to identify associated risk factors. If wonder if it is possible to calculate the prevalence of coagulation abnormalities in the given cohort of 70 consecutive children with CHD. Please give information about the statistic background.

2. Characteristics of CHD: Obviously, the definition of acyanotic and cyanotic CHD was given by diagnosis and not by saturations per se. How many patients in the acyanotic group had saturations below 90% , how many had severe pulmonary hypertension with Eisenmenger and were bad or no candidates for corrective surgery?

3. In the group of cyanotic CHD, patients with univentricular hearts which can be only palliated by Fontan Circulation are also included. Corrective surgery is not applicable for those.

4. Results: The statistics can be summarized and tables can be moved to the appendix. (Table 5, Table 6, table 8.

5. Discussion and conclusion: The discussion must be more coherend and shorter.. Due to the wide range of age some of the patients with ACHD are expected to have pulmonary hypertension and Eisenmenger which is known to result in severe coagulation abnormalities. This has to be worked up and discussed. That population is different to the population with CCHD and severe cyanosis, but without pulmonary arterial hypertension. Furthermore, there is a lack of therapeutic concepts that justify regular routine laboratory tests in patients, as the coagulopathies are a consequence of non-operated congenital heart disease. The only possible therapy is either corrective or palliative surgery to improve oxygenation (in CCHD) or prevent Eisenmenger as well as prevention of iron deficiency anemia. Hematological data must be available from the complete blood count. The data can be incorporate in the study or at least discussed. This is a major draw-back of the current version of the article and cannot be justified by the data.

6. Limitations: The number of patients is limited to calculate a prevalence.

This scientific work is very interesting and well done, but it does not justify the conclusion, I would recommend sticking only to the descriptive description

Reviewer #3: The authors present an interesting study regarding Coagulation abnormalities in a cohort of 70 children with uncorrected CHD in a developing country. They compared the results with controls.

They show that uncorrected CHD displays higher probability of coagulopathy, and among CHDs, Cyanotic ones are more likely going to develop such abnormalities.

Paper is well written, although there are minor spell mistakes to ammend.

- Please specify that patients were not on anticoagulant or antiplatelets other than aspirin

- Specify also other kind of medication patients were on, because many drugs can influence coagulation cascade

- Did you perform or are you willing to perform thromboelastogram to assess platelets funcion? In fact more than the number of platelets, is the activity which counts.

- Do you have data on liver function?

6. PLOS authors have the option to publish the peer review history of their article (what does this mean?). If published, this will include your full peer review and any attached files.

Reviewer #1: No

Reviewer #2: **Yes: **Ulrike Herberg

Reviewer #3: No

---

## [Author Response · Author response to Decision Letter 0]

22 Apr 2022

I appreciate all the comments from the Reviewers as they have helped to enrich the manuscript.

Reviewer #1: 1. This manuscript reviews 70 cases of uncorrected congenital heart disease in patients 6 months to 17 years of age to establish the prevalence of coagulation defects looking at very basic parameters including CBC, PT, PTT and D-dimer. The study found a significantly elevated incidence of 37% when compared to a control population of 7% abnormalities.

2. The case:control ratio is only 1:1 which limits the reliability of the results, particularly in view of the fact that the controls had a relatively high and unexpected incidence of coagulation abnormalities. A ratio of at least 1:2 would be preferred.

RESPONSE: Thank you very much for the comments. At the commencement of the comparative study, ratio of 1:1 was done to appropriately match the children for age and sex. The finding in healthy controls was unexpected and may be a limitation of the study.

3. The abbreviations CCHD and ACHD are used early in the article and not defined. It becomes apparent with further reading that it does refer to acyanotic and cyanotic heart disease, but it would be less confusion to define this earlier. 

RESPONSE: The comments are duly noted and amendments made.

4. The details of how the blood samples on pages 9 and 10 were collected are somewhat excessive and could be condensed considerably. It is important to stress the different amount of coagulant required for patients with > 55% hematocrit.

RESPONSE: Thank you for the comment. The details of the blood sample collection has been modified and the amount of anticoagulant added for patients with >55% hematocrit, the formula, has been included.

5. There is variability in referring to coagulation abnormalities in the singular or pleural form and should be uniformly made in the pleural form, that is "abnormalities".

RESPONSE: The comments are noted and have been corrected.

6. Many previous studies have shown that the major problems with coagulation occur with cyanotic congenital heart disease. This seems to be supported in this review which shows cyanotic lesions had a 37% incidence of abnormalities compared to a cyanotic lesions of 17%. However, most of the data discussion lumps acyanotic and cyanotic heart disease together and it would be important, even though the numbers are relatively small to break down the difference in these groups more clearly particularly in Table 1,3 and 4. It would also be important to include the pulse ox values in the acyanotic and cyanotic heart disease groups in addition to controls on Table 1. Including specific pulse ox data is critical since this is inherent in their conclusions of hypoxia being a primary factor in producing coagulation abnormalities.

RESPONSE: The data discussion has been separated as requested. The pulse oximetry has also been added to Tables 1 and 4.

7. In the discussion the authors describe the abnormalities of coagulation to 2 major factors, hypoxia and congestive heart failure. This speculation is intuitively correct, but should be expanded in more detail to include reference to low-grade DIC that is likely present with cyanotic patients. The link between congestive heart failure and probable hepatic hypoproduction of clotting factors should also be more clearly discussed.

RESPONSE: The discussion on hypoxia and congestive heart failure with reference to low grade/sub-clinical DIC has been expanded. So also, the link between congestive heart failure and hepatic production of clotting factors.

Reviewer #2: Thank you for the opportunity to review this interesting paper about coagulation disorders in patients with congenital, native heart defects from Nigeria. The paper is very well written and it is a pleasure to read the paper.

RESPONSE: Thank you so much for the comments.

However, there are some major concerns which have to be addressed:

1. The objective of the study was to determine the prevalence of coagulation abnormality in paediatric uncorrected CHD and to identify associated risk factors. If wonder if it is possible to calculate the prevalence of coagulation abnormalities in the given cohort of 70 consecutive children with CHD. Please give information about the statistic background.

RESPONSE: The static background (sample size calculation for difference in two proportion) has been added to the methodology.

2. Characteristics of CHD: Obviously, the definition of acyanotic and cyanotic CHD was given by diagnosis and not by saturations per se. How many patients in the acyanotic group had saturations below 90% , how many had severe pulmonary hypertension with Eisenmenger and were bad or no candidates for corrective surgery? 

RESPONSE: The comments are duly noted. Seven patients with ACHD had oxygen saturation less than 90%. One patient with ACHD had severe pulmonary hypertension. 

3. In the group of cyanotic CHD, patients with univentricular hearts which can be only palliated by Fontan Circulation are also included. Corrective surgery is not applicable for those.

RESPONSE: Comments noted with thanks. Such group of patients can only have palliative surgeries and the correction has been made.

4. Results: The statistics can be summarized and tables can be moved to the appendix. (Table 5, Table 6, table 8.

RESPONSE: Comments noted and effected.

5. Discussion and conclusion: The discussion must be more coherent and shorter. Due to the wide range of age some of the patients with ACHD are expected to have pulmonary hypertension and Eisenmenger which is known to result in severe coagulation abnormalities. This has to be worked up and discussed. That population is different to the population with CCHD and severe cyanosis, but without pulmonary arterial hypertension. Furthermore, there is a lack of therapeutic concepts that justify regular routine laboratory tests in patients, as the coagulopathies are a consequence of non-operated congenital heart disease. The only possible therapy is either corrective or palliative surgery to improve oxygenation (in CCHD) or prevent Eisenmenger as well as prevention of iron deficiency anemia. Hematological data must be available from the complete blood count. The data can be incorporate in the study or at least discussed. This is a major draw-back of the current version of the article and cannot be justified by the data.

RESPONSE: Thank you for the comments. The proportion of ACHD with severe pulmonary hypertension has been discussed. The recommendation on therapy is also noted. Iron deficiency and other hematological indices were not evaluated in this study, as the study focused on coagulation abnormalities. It is however a possibility for future research.

6. Limitations: The number of patients is limited to calculate a prevalence.

This scientific work is very interesting and well done, but it does not justify the conclusion, I would recommend sticking only to the descriptive description

RESPONSE: Thank you for the comments. The recommendations are duly noted.

Reviewer #3: The authors present an interesting study regarding Coagulation abnormalities in a cohort of 70 children with uncorrected CHD in a developing country. They compared the results with controls.

They show that uncorrected CHD displays higher probability of coagulopathy, and among CHDs, Cyanotic ones are more likely going to develop such abnormalities.

Paper is well written, although there are minor spell mistakes to amend.

RESPONSE: Thank you for the comments.

- Please specify that patients were not on anticoagulant or antiplatelets other than aspirin.

RESPONSE: The patients were not on anticoagulants nor antiplatelets. This has been included in the methodology.

- Specify also other kind of medication patients were on, because many drugs can influence coagulation cascade 

RESPONSE: The patients were on frusemide, spironolactone and propranolol. This has been included in the methodology.

- Did you perform or are you willing to perform thromboelastogram to assess platelets funcion? In fact more than the number of platelets is the activity which counts. 

RESPONSE: Thank you for the comment. Only platelet count was evaluated in this study. Thromboelastogram is not available in our environment, and was stated as a limitation to the current study. It is however noted as a possible area of future research.

- Do you have data on liver function? 

RESPONSE: Liver function test was not performed in this study.

---

## [Editor Report · Decision Letter 1]

1 May 2022

Coagulation abnormalities in children with uncorrected congenital heart defects seen at a teaching hospital in a developing country

PONE-D-22-02924R1

Dear Dr. Omotola O Majiyagbe,

We’re pleased to inform you that your manuscript has been judged scientifically suitable for publication and will be formally accepted for publication once it meets all outstanding technical requirements.

Kind regards,

Emanuele Bobbio, M.D.

Section Editor

PLOS ONE

Additional Editor Comments:

Congratulations on your manuscript which is very well written and further improved.
---

## [Editor Report · Acceptance letter]

30 May 2022

PONE-D-22-02924R1 

Coagulation abnormalities in children with uncorrected congenital heart defects seen at a teaching hospital in a developing country.

Dear Dr. Majiyagbe:

I'm pleased to inform you that your manuscript has been deemed suitable for publication in PLOS ONE. Congratulations! Your manuscript is now with our production department. 

Kind regards, 

on behalf of

Dr. Emanuele Bobbio 

Section Editor

PLOS ONE